# The Use of Pro-Angiogenic and/or Pro-Hypoxic miRNAs as Tools to Monitor Patients with Diffuse Gliomas

**DOI:** 10.3390/ijms23116042

**Published:** 2022-05-27

**Authors:** Guénaëlle Levallet, Fatéméh Dubois, Arthur Leclerc, Edwige Petit, Lien Bekaert, Maxime Faisant, Christian Creveuil, Evelyne Emery, Gérard Zalcman, Emmanuèle Lechapt-Zalcman

**Affiliations:** 1Normandie University, UNICAEN, CNRS, ISTCT Unit, GIP CYCERON, F-14074 Caen, France; fatemeh.dubois@unicaen.fr (F.D.); epetit@cyceron.fr (E.P.); emmanuele.lechapt@aphp.fr (E.L.-Z.); 2Department of Pathology, CHU de Caen, F-14033 Caen, France; faisant-m@chu-caen.fr; 3Department of Neurosurgery, CHU de Caen, F-14033 Caen, France; arthur.leclerc@neurochirurgie.fr (A.L.); lien.bekaert@gmail.com (L.B.); emery-e@chu-caen.fr (E.E.); 4Biomedical Research Unit, CHU de Caen, F-14033 Caen, France; creveuil-cr@chu-caen.fr; 5Department of Thoracic Oncology & CIC1425, Paris Cité University, Hôpital Bichat-Claude Bernard, Assistance Publique Hôpitaux de Paris, F-75877 Paris, France; gerard.zalcman@aphp.fr; 6U830 INSERM “Cancer, Heterogeneity Instability and Plasticity (CHIP), STRESS Group”, Curie Institute Research Center, F-75005 Paris, France; 7Department of Pathology, GHU AP-HP University Hospital Henri-Mondor, F-94010 Creteil, France; 8INSERM U955 Institut Mondor de Recherche Biomédicale (IMRB), Paris Est Créteil University, F-94010 Creteil, France

**Keywords:** miRNA, hypoxia, angiogenesis, glioma

## Abstract

IDH (isocitrate dehydrogenase) mutation, hypoxia, and neo-angiogenesis, three hallmarks of diffuse gliomas, modulate the expression of small non-coding RNAs (miRNA). In this paper, we tested whether pro-angiogenic and/or pro-hypoxic miRNAs could be used to monitor patients with glioma. The miRNAs were extracted from tumoral surgical specimens embedded in the paraffin of 97 patients with diffuse gliomas and, for 7 patients, from a blood sample too. The expression of 10 pro-angiogenic and/or pro-hypoxic miRNAs was assayed by qRT-PCR and normalized to the miRNA expression of non-tumoral brain tissues. We confirmed in vitro that IDH in hypoxia (1% O_2_, 24 h) alters pro-angiogenic and/or pro-hypoxic miRNA expression in HBT-14 (U-87 MG) cells. Then, we reported that the expression of these miRNAs is (i) strongly affected in patients with glioma compared to that in a non-tumoral brain; (ii) correlated with the histology/grade of glioma according to the 2016 WHO classification; and (iii) predicts the overall and/or progression-free survival of patients with glioma in univariate but not in a multivariate analysis after adjusting for sex, age at diagnosis, and WHO classification. Finally, the expression of miRNAs was found to be the same between the plasma and glial tumor of the same patient. This study highlights a panel of seven pro-angiogenic and/or pro-hypoxic miRNAs as a potential tool for monitoring patients with glioma.

## 1. Introduction

Adult diffuse gliomas are the most common primary malignant brain tumors, accounting for approximately 60% of all central nervous system tumors. These tumors are characterized by a number of criteria, either morphological/histological (tumoral cells mitoses, microvascular proliferation, hypoxia/necrosis as infiltration (for review: [1])) or molecular (isocitrate dehydrogenase (IDH) mutations, 1p/19q codeletion, ATRX (alpha thalassemia/mental retardation syndrome X-linked) mutations, mutations in the promoter of *TERT* (telomerase reverse transcriptase), etc., [2,3,4]). Since the revision of the classification of brain tumors according to the World Health Organization (WHO) in 2016 [2], which was reviewed in 2021 [5], all these morphological/histological and molecular criteria have been integrated by pathologists to establish a histoprognostic grade of these tumors for each histological diagnosis. Some molecular abnormalities (for example, trisomy 7 associated with monosomy 10, mutation of the *TERT* promoter, biallelic deletion of the *CDK2NA* gene) alone allow us to classify the tumor into grade 4, independently of the histomorphological criteria [5].

Diffuse gliomas in adults are therefore now well characterized from a molecular point of view; however, there is still no molecular tool for monitoring these patients longitudinally, even though a low-grade mutated *IDH* tumor will invariably evolve into a higher-grade tumor [6]. It is therefore necessary to improve the care and follow-up of patients with diffuse gliomas by identifying such tools. It is acknowledged that the occurrence of the *IDH1/2* mutation remains the upstream genetic event in two diffuse glioma lineages: diffuse astrocytomas and oligodendrogliomas [3,4]. It is also known that the *IDH1/2* mutation, hypoxia leading to necrosis, and microvascular proliferation are interrelated [1,2,7,8]. *IDH1/2* mutation can control these phenomena by modulating the expression of miRNAs. Indeed, *IDH1/2* mutation is related to epigenetic modification by both DNA hypermethylation [9,10] and a change in miRNA (miRNA) expression (for review [8]). Interestingly, in lower-grade gliomas, the *IDH1/2* mutation has more of an impact on miRNA expression than histological and other genomic features [11].

By repressing transcription or inducing the degradation of their target mRNA molecules, miRNAs can control cell growth, proliferation, metabolism, and apoptosis [12]. Many miRNAs are dysregulated in gliomas and are linked to their development and progression [13,14] (for review: [15]). Among these, some appear oncogenic, such as tumor suppressor miRNAs [16] as well as miRNAs correlated with the grade and/or histology of the glial tumor and/or with the outcome of glioma patients, as reviewed in a recent meta-analysis based on the data of 4708 glioma patients [17]. For example, a high expression of miR-15b, 21, 148a, 196, 210, and 221 or a low expression of miR-106a and 124 predicts a poor prognosis in glioma patients, while the expression of miR-10b, 17, 20a, 155, 182, 200b, and 222 fails to predict such survival [17]. Among these miRNAs, some are induced by hypoxia and/or neoangiogenesis, the two hallmarks of glioma history [1], such as mir210 [18,19]. Interestingly, the detection of stable miRNA expression in cerebrospinal fluid, blood serum, and other bodily fluids has led to the possibility of using miRNAs as non-invasive biomarkers for clinical applications [20,21]. However, specific miRNAs still need to be elucidated in the diagnosis of a glioma, especially in the early screening stage [22].

As IDH genes play important roles in the mechanism of glioma, here, we tested the diagnostic and prognostic values of miRNAs, reflecting on the features of gliomas in the WHO 2016 classification and patients’ survival in a series of 97 grade II to IV gliomas. We focused on ten miRNAs: has-mir-200b-3p, -200c-3p, -210-3p, -100-5p, -126-5p, -132-3p, -221-3p, -424-5p, -128-3p, and -451-5p. These are miRNAs which could be involved in the regulation of hypoxia/cell proliferation/differentiation of glioma cells, as described for other cell types [23,24,25,26,27,28,29,30,31,32,33] (for review: [15,34]).

## 2. Results

### 2.1. Expression of the Pro-Angiogenic and/or Pro-Hypoxic miRNAs Studied Is Affected by IDH Mutation and Hypoxia In Vitro

We first validated the influence of IDH1 mutation on proangiogenic miRNA expression by comparing the expression of these miRNAs between two isogenic HBT-14 (U-87 MG) cell lines only differing in their expression of either wild-type IDH1 (HBT-14 (U-87 MG) *IDH1^WT^*) or mutated IDH1 (HBT-14 (U-87 MG) *IDH1^R132H^*) (Figure 1 and Figure 2). We report that four of the miRNAs studied had significantly altered expressions (has-mir-100-5p, 128-3p, -221-3p and -451-5p) when the cell line expressed the IDH1 R132H-mutant, while the others showed an expression variation that did not reach significance, although some looked to be substantial (for example, has-mir-210-3p and -424-5p). To note, the mir-126-5p was undetectable in these lines.

Microvascular proliferation is linked to hypoxia, a hallmark of high-grade gliomas, which is itself counterbalanced by this microvascular proliferation, although the neo-vessels formed are defective [1]. Thus, we next tested the influence of hypoxia by incubating HBT-14 (U-87 MG) *IDH1^WT^* and HBT-14 (U-87 MG) *IDH1^R132H^* cells for 24 h under 1% O_2_. Morphologically, we observed that the cells survived in these growing conditions, although they appeared less numerous than their homologues cultivated in physoxia, showed a more star-shaped form, and featured more interconnections (Figure 2).

The quantification of miRNAs from HBT-14 (U-87 MG) *IDH1^WT^* and HBT-14 (U-87 MG) *IDH1^R132H^* grown in normoxia or hypoxia (1% O_2_, 24 h) revealed that hypoxia could interfere with the IDH1 mutation and influence miRNA expression when compared to cells grown in normoxia (Figure 3).

We measured significant increases in mir-210-3p and mir-424-5p in both HBT-14 (U-87 MG) *IDH1*^WT^ and HBT-14 (U-87 MG) *IDH1*^R132H^ when grown for 24h under 1% O_2_ (Figure 3). We also reported significant decreases in mir-128-3p and mir-221-5p in HBT-14 (U-87 MG) *IDH1*^WT^ grown for 24h under 1% O_2_ (Figure 3), but not in IDH1-mutant cells. Therefore, hypoxia modified the expression of mir-210-3p and mir-424-5p studied here in HTB-14 (U-87 MG) cells regardless of *IDH1/2* status, unlike the mir-128-3p and mir-221-5p whose variation in expression under hypoxia seems to be linked by the presence of a wild-type IDH.

### 2.2. The Expression of Pro-Angiogenic and/or Pro-Hypoxic miRNAs Is Strongly Affected in Patients with Glioma

The characteristics, treatment history, and pathologic data pertaining to 97 glioma samples from the 97 patients studied are summarized in Appendix A. The median age was 53.9 years [range: 22.3–79.3]. There were 37 females and 60 males. The median follow-up period was 28.85 months [range: 0.26–304.23 months]. According to the 2016 WHO classification [2], the 97 glioma samples were classified as follows: 11 Grade II, isocitrate dehydrogenase 1/2 (IDH)-mutant and 1p19q-codeleted oligodendrogliomas (O); 16 Grade III, IDH-mutant and 1p19q-codeleted anaplastic oligodendrogliomas (AO); 18 Grade II, diffuse and IDH-mutant astrocytomas (A-IDH^MUT^); 7 Grade III, IDH-mutant anaplastic astrocytomas (AA-IDH^MUT^); 8 Grade IV, IDH-mutant glioblastomas (GB-IDH^MUT^); 37 Grade IV, IDH-wild-type glioblastomas (GB-IDH^WT^) (Appendix A).

We first assessed the level of expression of has-miR-200b-3p, -200c-3p, -210-3p, -100-5p, -126-5p, -132-3p, -221-3p, -424-5p, -128-3p, and -451-5p between glial tumors regardless of their WHO classification. We observed that the expression of each miRNA was modified by at least >2 fold (increase or decrease) as compared with non-tumoral brain tissue in 62.8 to 84.5% of patients with glioma according to the concerned miRNA (Figure 4).

We report that three miRNAs (mir-221-3p, mir-132-3p, and mir-128-3p, histograms in red in Figure 4) were decreased in glioma specimens compared to normal brain tissue, while the seven other miRNAs studied were all increased (histograms in green in Figure 4). We also observed that two miRNAs (mir-210-3p and mir-451-5p) varied markedly between different subtypes of glial tumors (Figure 4). In the same patient’s tumor specimen, the changes in miRNAs expression seen were not exclusively increases or decreases; some miRNAs increased when the others decreased. There was also no change in miRNA exclusive to the other miRNAs which was expected since these miRNAs reflect interrelated features in gliomas. We even report that, as detailed in Table 1, these miRNA expressions appeared to be strongly correlated with each other, except for mir-128-3p and mir-210-3p.

Indeed, as seen in Table 1, the Spearman rank correlation coefficient testing the interaction between each pair of miRNAs varied from 0.344 to 0.822 and was strongly significant (*p* < 0.001) except for mir-128-3p and mir-210-3p (Spearman rank correlation coefficient: 0.078, *p* = 0.45).

### 2.3. The Expression of Pro-Angiogenic and/or Pro-Hypoxic miRNAs Is Correlated with the 2016 WHO Classification

Having observed that the expression of microRNAs was affected in patients with glial tumors (Figure 4), we next investigated which feature of glioma could have influenced the expression of the miRNAs studied here. We thus looked at the miRNA expression according to (1) the mutational status of IDH (Table 2), (2) the microvascular proliferation (Table 3), and (3) the histological subgroup (Table 4).

As shown in Table 2, 7/10 miRNAs studied here were more strongly expressed in glioma *IDH1/2*-^WT^ (expression > 100) than in non-tumor brain tissue, while only 2/10 miRNAs were more strongly expressed in glioma *IDH1/2*-^MUT^ than in non-tumor brain tissue. Thus, the *IDH1/2* mutation significantly decreased the expression of the miRNAs studied here by 4.2 to 8.8-fold according to the miRNA and except for mir-128-3p (increased by *IDH1/2* mutation) and mir-100-5p (decrease not significant).

As shown in Table 3, we further report that the microvascular proliferation also significantly increased the expression of pro-angiogenic or pro-hypoxic miRNAs up to 4.3 folds (except for mir-128-3p, mir-132-3p, and mir-100-5p (not significant). Thus, the expression of 7/10 miRNAs studied here was stronger when the glial tumor exhibited a microvascular proliferation than when it did not.

Finally, as shown in Table 4, the histological subgroup influenced the expression of the pro-angiogenic or pro-hypoxic miRNAs studied here. Indeed, the level of expression of the microRNAs mir-100-5p, -126-3p, -128-3p, -132-3p, -210-5p, and -221-3p was comparable between astrocytic tumors (A, AA) and oligodendroglial tumors (O, AO).

Conversely, the mir-200b-3p was found to be more expressed in astrocytic (A, AA) than in oligodendroglial tumors (O, AO). Similarly, the expression of mir-451-5p increased between O and AO, but not between A and AA. Other miRNAs could be used for comparison: the expression of mir-200c-3p, -424-5p, and -451-5p increased with the tumor evolution (A > AA, O > AO).

### 2.4. The Expression of Pro-Angiogenic and/or Pro-Hypoxic miRNAs Predicts Overall (OS) and Progression-Free (PFS) Survival in Patients with Glioma in Univariate Analysis

We next tested the influence of the expression of pro-angiogenic and/or pro-hypoxic miRNAs on the OS and PFS using univariate and multivariate Cox-proportional hazard models. As detailed in Table 5, except for mir-128-3p, mir-132-3p, and mir-100-5p, the expression of the miRNAs studied here predicted the OS and PFS of patients with glioma in the univariate analysis (*p* < 0.001) but not in the multivariate analysis, following adjustments for sex, age at diagnosis, and WHO 2016 classification.

### 2.5. Mir-128-3p Predicts a Poorer PFS in Patients with AA-IDH^MUT^ or AO-IDH^MUT^ and mir-100-5p Predicts a Poorer PFS in Patients with AA-IDH^MUT^

For each miRNA, we used an interaction test to compare the hazard ratios between the WHO 2016 classes for OS and PFS. Interactions were not statistically significant for OS, but for PFS, significant differences were found for mir-128-3p and mir-100-5p (*p* = 0.033 and *p* = 0.013, respectively; Appendix A).

We thus further calculated the prognostic value of these two miRNAs for each WHO subgroup and reported that mir-128-3p predicts a poorer PFS in patients with AA-IDH^MUT^ or AO-IDH^MUT^ and that mir-100-5p predicts a poorer PFS in patients with AA-IDH^MUT^ (Table 6). However, the low number of patients by class (sometimes less than 10 subjects) means that it is necessary to interpret such results with caution.

### 2.6. For the Same Patient, Expression of Plasma miRNAs Coincides with the Expression of Tumoral miRNAs

We know that stable miRNA expression is measurable from bodily fluids such as cerebrospinal fluid or blood serum [20,21] but that specific miRNAs in the diagnosis of a glioma, in particular at the early stage, are still missing [22]. We thus next tested the possibility of directly assaying the microRNAs studied in this study from the patient’s blood sample.

A blood sample was collected from seven patients, from which we also extracted and quantified the pro-angiogenic miRNAs. All the miRNAs could be assayed from these blood samples (no amplification failure and the number of CTs after real-time PCR amplification was <35), even if, for some miRNAs in the circulatory system, their expression patterns were at a slightly lower concentration compared to the tumoral tissues. As illustrated in Figure 5 for four miRNAs (has-miR-100-5p, -132-3p, -200b-3p, and -221-3p), the amount of each miRNA detected in the glial tumor was correlated with that detected in the blood of the same patient, except for the miR-100-5p in patient 3.

## 3. Discussion

It is now well known that miRNAs are involved in tumor initiation and development [35]. Such miRNAs thus appear to be interesting tools for use in the diagnosis and/or monitoring of cancers, whether in gliomas or other tumors, particularly due to their non-invasive nature since they can be measured from a patient’s blood sample. Indeed, the expression of miRNAs in blood and tissues has tumor-related and tissue-specific features, and their expression is remarkably stable [36]. Nevertheless, miRNAs have not yet come to be used in daily clinical practice. To relaunch the debate on their possible clinical interest, particularly for the diagnosis of patients with gliomas, a meta-analysis recently tested the diagnostic performance of circulating miRNAs for gliomas [22]. After analyzing 18 articles covering 24 studies containing 2170 glioma patients and 1456 healthy participants, the authors concluded that circulating miRNAs have the potential to serve as diagnostic biomarkers for gliomas. Our results are in agreement with their conclusion; indeed, we prove here that we can use miRNAs as tools for monitoring patients with gliomas by selecting several miRNAs involved in intertwined phenomena and accounting for the natural history of the disease. Indeed, we tested the diagnostic and prognostic values of miRNAs reflecting the features of gliomas (IDH mutation, microvascular proliferation, hypoxia) and patients’ survival in a series of 97 gliomas of grades II to IV to determine whether miRNAs could be used as tools for monitoring patients with gliomas. As detailed in the Section 1, we thus chose the miRNAs involved in the regulation of hypoxia/cell proliferation/differentiation of glioma cells, as described for other cell types [23,24,25,26,27,28,29,30,31,32,33] (for review: [15,34]). As expected, since the miRNAs studied here reflected the mechanisms related to gliomagenesis, these miRNAs appeared to be strongly correlated with each other, except for mir-128-3p and mir-210-3p.

First, we successively reported that the expression of some of the pro-angiogenic and/or pro-hypoxic miRNAs studied could actually be affected by IDH mutation and hypoxia in vitro, and then in tumor specimens from patients with glioma that the expression of some pro-angiogenic and/or pro-hypoxic miRNAs could be strongly affected by and correlated with the 2016 WHO classification (*IDH1/2* mutation, microvascular proliferation, histoprognostic group according to WHO 2016 classification). Interestingly, the induced hypoxia of IDH HTB-14 (U-87 MG) cells modified the expression of four miRNAs, for two (mir-210-3p and mir-424-5p, which decreased under hypoxic conditions) independently of the IDH mutation, while for the two others (128-3p and mir-221-5p, which decreased under hypoxic conditions), only in wild-type IDH HTB-14 (U-87 MG) cells. Such results could explain why IDHWT glioma are more aggressive tumors with a higher microvascular proliferation than IDHMUT glioma; indeed, the mir-128-3p is a tumor suppressor [37] and both the mir-128-3p and the mir-221-5p sometimes display anti-angiogenic behavior by targeting, in particular, VEGFC [38] and the Hypoxia-inducible factor 1 alpha, respectively [39]. Then, in a small sample of seven patients for whom both tissue and blood were available, we then reported that the expression of pro-angiogenic and/or pro-hypoxic miRNAs in plasma coincided with the expression of tumor miRNAs and that the tumor expression of some miRNAs could predict OS and PFS in patients with glioma, at least in a univariate analysis. We thus suggest the use of pro-angiogenic and/or pro-hypoxic miRNAs as tools for monitoring patients specifically with *IDH1/2*-mutated gliomas. Indeed, we also observed in this study that the variations in the expression of miRNAs within the group of patients with GB IDH WT was specific to this group, which is consistent with the report that miRNA profiles play a more significant prognostic role in *IDH*-mutant tumors than in *IDH* wild-type tumors [11,40]. Moreover, the *IDH1/2* mutation status had a greater impact than the histological and other genomic features on miRNA expression patterns; 361/487 (74%) of the miRNAs were differentially expressed according to their *IDH1/2* mutation status [11].

Our results, which indicate that we could use miRNAs as tools for monitoring patients with *IDH1/2*-mutated gliomas, are consistent with the study of Chen and co-workers, based on another miRNA, mir-720, assayed on 122 patients with glioma (Stage I: 20; Stage II: 17; Stage III: 35; and Stage IV: 50, according to the WHO 2016 classification). Chen an al. reported that the plasma miR-720 was associated with the tumor grade and associated with recurrence or development in patients with glioma but that the sensitivity and specificity results indicated that the diagnostic ability of miR-720 for glioma was only moderate [41]. This result raises the question of what methodological choices should be made when one wants to use miRNAs as diagnostic and/or monitoring tools for patients with tumors and more particularly gliomas. Although circulating miRNAs are promising diagnostic biomarkers for patients with glioma, the serum miRNAs and miRNA panels presented a superior diagnostic performance compared to the use of only one miRNA [22]. Moreover, variations in the plasma concentrations of only one miRNA could alternatively reflect another cancer pathology in the same patient and not be specific to glioma, thus compromising the potential clinical utility in diagnosis and follow-up by leading to false detection. To overcome these possible biases, it is therefore more relevant to assay a signature of miRNAs—moreover, a signature of miRNAs reflecting the specific characteristics of the tumor that one seeks to characterize. As an example, the use of a four-miRNA risk classifier (miR-10b, miR-130b, miR-1304, and miR-302b), involved in the proliferation, invasion, and survival of glioma or other tumors cells [42,43,44,45], would allow one to independently distinguish cases as either at a high or low risk of poor prognosis in *IDH1/2*-mut lower-grade glioma [11].

Finally, the fact that the plasma dosage of miRNAs could reflect the tumor dosage is also consistent with what other authors have been able to report in the literature, such as during Spinal Cord Glioma Progression [46]. Indeed, the brain is among the tissues with the strongest correlation for microRNA for both plasma and serum [47].

In our work, mir-128-3p, mir-132-3p, and mir-100-5p often behaved differently from the other miRNAs that we chose to study; these miRNAs were not linked to the WHO 2016 classification and did not predict the OS and PFS of patients with glioma in univariate analysis. Conversely, mir-128-3p predicted a poorer PFS in patients with AA-IDH^MUT^ or AO-IDH^MUT^, while mir-100-5p predicted a worse PFS in patients with AA-IDH^MUT^. This, again, is consistent with the tumor suppressor role attributed to mir-128-3p [37], which we confirmed by identifying a drop in its expression compared to the level quantified in non-tumor brain tissue, and with the work of Zhang et al., 2019, who reported that a low miR-100 expression correlated with worse clinicopathological characteristics such as Karnofsky Performance Scale and *IDH1/2* mutation status [48]. It remains difficult to explain why these three miRNAs behave differently: it is possible that the variations in such miRNAs are more difficult to highlight since they are constitutively highly expressed in the brain [49,50] and that, regardless of the processes in which they participate, their variations are masked and under-evaluated in this tissue. In any case, it would be preferable to exclude these miRNAs from the panel of miRNAs used for characterizing angiogenesis and for hypoxia to be analyzed to evaluate the progression of the disease in patients with *IDH*-mutated gliomas.

## 4. Materials and Methods

### 4.1. Tissue Samples and Patient Characteristics

Ninety-seven patients, aged 18 years or older with a tissue diagnosis of WHO grade II, III, and IV diffuse gliomas made between September 2001 and March 2012, were identified from the brain tumor registry of Caen University Hospital, France. Characteristics, treatment history, and pathologic data from these patients are summarized in Appendix A. For 7 patients, a blood sample was also available. The 8 non-tumoral brain tissues were from patients without glioma who underwent brain surgery.

All patients provided informed consent regarding the collection of tumor specimens and their molecular evaluation, as required by French law. The study was approved by the institutional ethics committee of Caen University Hospital, France (DC-2008-588). All tumor specimens were reviewed by a neuropathologist (ELZ) to confirm the diagnosis and grade according to the new classification system adopted by the World Health Organization (WHO) in 2016, as described in [10]. Indeed, we used the classification in place at the time of the constitution of this cohort, i.e., the WHO 2016 classification and could not update on the classification of 2021 due to the exhaustion of numerous tumor specimens.

### 4.2. Cell Culture and Hypoxia Treatment

The human glioblastoma cell lines HBT-14 (U-87 MG) *IDH1^WT^* and HBT-14 (U-87 MG) *IDH1^R132H^* cells obtained from the generous gift of Pr. Marc Sanson (Hospital Group Pitié-Salpêtrière, Paris, France) [51], were maintained in Dulbecco’s modified Eagle’s medium (DMEM) supplemented with 10% fetal bovine serum (FBS) and 1% penicillin/streptomycin. Normoxic cells (21% O_2_) were grown in a humidified air atmosphere incubator containing 95% air/5% CO_2_ at 37 °C. Hypoxia experiments were performed in a controlled atmosphere chamber (INVIVO2 1000, Ruskinn, Awel, France) set at 1% O_2_, 94% N_2_, and 5% CO_2_ at 37 °C for 24 h.

### 4.3. DNA and miRNA Extraction

DNA from HBT-14 (U-87 MG) cells was extracted using the QIAmp DNA kit (Qiagen™) according to the manufacturer’s recommendation.

The miRNAs from HBT-14 (U-87 MG) cells were extracted using miRNAeasy (Qiagen™). The miRNAs from the 97 FFPE (formalin-fixed paraffin-embedded) surgically resected tumor specimens or the 8 non-tumoral brain tissues were extracted on 3 adjacent 15 μm cuts using the miRNAeasy-FFPE kit (Qiagen™, Hilden, Germany). For the tumoral specimens, morphological control was systematically carried out beforehand by a neuropathologist (ELZ) in order to guarantee that the percentage of tumor cells was greater than 70% and the absence of areas of necrosis or hemorrhage. When this was not the case, a macro-dissection of the samples was performed to determine these quality criteria. miRNAs were extracted from the seven plasma samples using NucleoSpin miRNA Plasma (Macherey-Nagel™, Düren, Luxembourg). For each sample, miRNA extraction was carried out according to the respective manufacturer’s instructions.

The integrity and quality of the purified DNA were assessed by 1% agarose gel electrophoresis, and the DNA/miRNA concentration was measured with the NanoDrop 2000 spectrophotometer (Thermo Fisher Scientific, Asnières-sur-Seine, France).

### 4.4. Quantitative Real-Time Reverse Transcription: PCR

The miRNAs were retrotranscribed and amplified (PCR) using the TaqMan MiRNA Reverse transcription kit (Applied Biosystem, Birchwood, United Kingdom). An amount of 5 ng of miRNA was used for every miRNA that we tested. MiRNA expression was analyzed using the following TaqMan MiRNA assays (Applied Biosystem Birchwood, United Kingdom): has-miR-200b-3p (Assay ID: 002251), has-miR-200c-3p (002300), has-miR-210-3p (000512), has-miR-100-5p (000437), has-miR-126-5p (000451), has-mir-132-3p (000457), hsa-miR-221-3p (000524), has-miR-424-5p (000604), has-miR-128-3p (002216), and has-miR-451-5p (001141) (targets for each miRNA are listed in Appendix A).

The RT-PCR data were normalized to the small nucleolar house-keeping RNA, RNAS RNU48 (SNORD48) (assay ID 001006). Positive standards and reaction mixtures lacking the reverse transcriptase were used routinely as controls for each miRNA sample. Relative quantification was conducted using the deltaCt method, where deltaCt is CtmiRX-CtRNU48. To facilitate comparison between conditions (histology, grade), the miRNAs were further normalized to the miRNA expression in healthy brain tissue (quantification averaged from the normal brain tissue of 8 subjects, operated on to cure their epilepsy). As presented in Appendix A, the delta-CT averages of each miRNA of these 8 patients were calculated along with the SD (standard deviation) and did not vary more than one CT between the brain tissue specimens of these patients.

Each miRNA was thus finally expressed in base 100, with the value 100 being attributed to the delta-CT of the miRNA measured in normal brain tissue.

### 4.5. IDH1 and IDH2 Mutations Assay

HBT-14 (U-87 MG) cells with wild-type IDH1 (HBT-14 (U-87 MG) *IDH1^WT^*) or mutated IDH1 (HBT-14 (U-87 MG) *IDH1^R132H^*) were certificated by *IDH1^R132H^* staining (Appendix A). Cells were collected, washed with PBS, and fixed in neutral buffered formalin for 20 min. After centrifugation, the pellet was embedded in 6–10 drops of melted Bio-Agar (Bio-Optica, Milano, Italy, cod. 05-98035) and chilled in the freezer (−20 °C) until complete solidification was achieved. Next, the sample pellet was putted between two pads of a bio-cassette and then processed and paraffin-embedded according to the histologic routine. For IDH1 (R132H) detection, 3 μm sections were cut and placed on poly-L-lysine-coated slides. Immunohistochemical staining was performed using the Ventana Discovery XT automated immunostainer (Ventana Medical Systems, Tucson, AZ, USA). Slides were subjected to deparaffinization in xylene and hydration through a series of decreasing alcohol concentrations, following standard procedures. Antigen retrieval was performed using a high-pH Tris-based solution (CC1; VMS) for 64 min at 100 °C. The slides were incubated with the anti-IDH1 R132H primary antibody (Diagomics, Blagnac, France, IHC132-100) at a 1:100 dilution and then with the Ventana UltraView detection kit and Ventana DAB. Counterstaining with hematoxylin was performed on the Leica ST 5020. The slides were finally washed in running water for 10 min, dehydrated, cleared, and mounted with resinous mounting medium.

### 4.6. Statistical Analyses

In vitro data are presented as means ± SEM (*n* ≥ 3). Statistical differences were determined by one-way analysis of variance (ANOVA) followed by Dunnett’s Multiple Comparison Test (GraphPad Software, Inc., San Diego, CA, USA). Statistical significance was set at *p* ≤ 0.05.

Spearman’s rank correlation coefficient was used to test the correlation between the miRNAs. Comparisons of the expression of miRNAs according to *IDH1/2* mutation, microvascular proliferation, and WHO2016 classification were made using the Mann–Whitney and Kruskal–Wallis tests. Univariate and multivariate Cox-proportional hazard models were used to assess the prognostic value of the miRNA expression. Hazard ratios (HR) were estimated with 95% confidence intervals (95%CI). For each miRNA, the hazard ratios were compared between the WHO 2016 classes by including an interaction term in the Cox models. Statistical significance was set at *p* < 0.05. The data were analyzed with the IBM SPSS software (New York, NY, USA), Version 22.

## 5. Conclusions

With this work, we reported that the expression of a panel of seven pro-angiogenic and/or pro-hypoxic miRNAs (has-miR-200b-3p, -200c-3p, -210-3p, -126-5p, -221-3p, -424-5p, and -451-5p) was affected in patients with glioma and related to the histology/grade of glioma according to the 2016 WHO classification. We also suggested that, by predicting glioma patients’ overall and/or progression-free survival in a univariate analysis, pro-angiogenic and/or pro-hypoxic miRNAs can be used as tools for monitoring patients, specifically with IDH-mutated low-grade tumors, since they are also measurable in plasma. However, our results remain preliminary because of the small sample size, the lack of a longitudinal follow-up for patients with plasma sampling, and the need to verify these results in prospective dedicated studies before being used for diagnosis, to monitor response to treatment, or to assess the risk of residual disease and relapse after surgical resection.

## Figures and Tables

**Figure 1 ijms-23-06042-f001:**
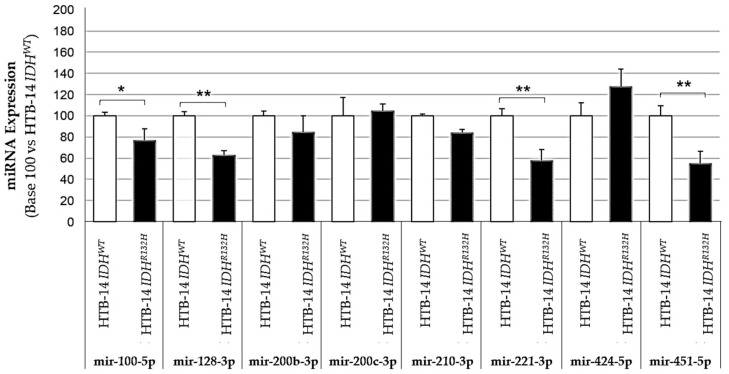
Expression of pro-angiogenic or pro-hypoxic miRNAs according to the presence (HBT-14 (U-87 MG) *IDH1*^R132H^ cells) or absence (HBT-14 (U-87 MG) *IDH1*^WT^ cells) of the IDH1 R132H mutation. The miRNAs were extracted from cell lines using miRNAeasy (Qiagen™), then retrotranscribed (RT) and amplified (PCR) using the TaqMan MiRNA Reverse transcription kit (Applied Biosystem). The RT-PCR data were normalized to the small nucleolar house-keeping RNA, RNA RNU48 (SNORD48) (assay ID 001006). Each miRNA was expressed in base 100 (100 being attributed to the delta-CT of the miRNA measured in HBT-14 (U-87 MG) *IDH1*^WT^ cells) (*n* = 3, ANOVA followed by a post hoc Dunnett’s test, *: *p* < 0.05, **: *p* < 0.01). The mir-126-5p was undetectable in these lines.

**Figure 2 ijms-23-06042-f002:**
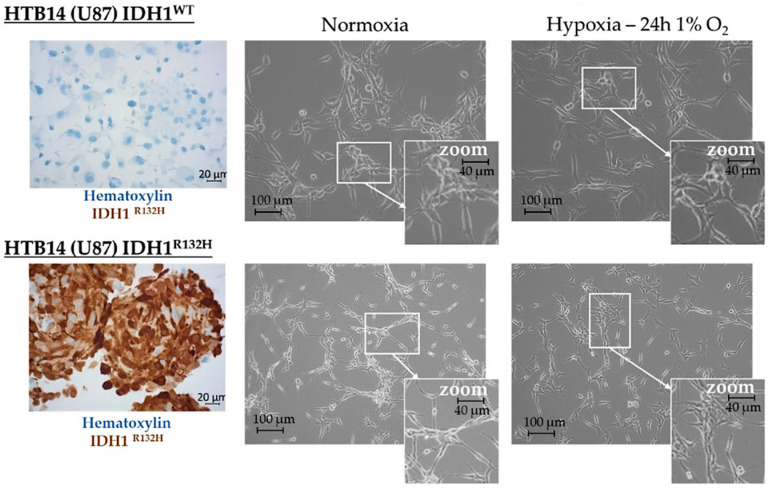
Morphological appearance of HBT-14 (U-87 MG) *IDH1^WT^* and HBT-14 (U-87 MG) *IDH1^R132H^* grown in normoxia or hypoxia (1% O_2_, 24 h). HBT-14 (U-87 MG) *IDH1^WT^* and HBT-14 (U-87 MG) *IDH1^R132H^* cells, as validated by immunohistochemical staining carried out according to standard procedures against *IDH1^R132H^* (left panel), reaching 60% confluence, were cultivated for an additional 24 h in physoxia or hypoxia (0.1% O_2_). The appearance of these cells was imaged under a phase contrast microscope: the right panel presents representative photos of these cells according to the culture condition (normoxia/hypoxia).

**Figure 3 ijms-23-06042-f003:**
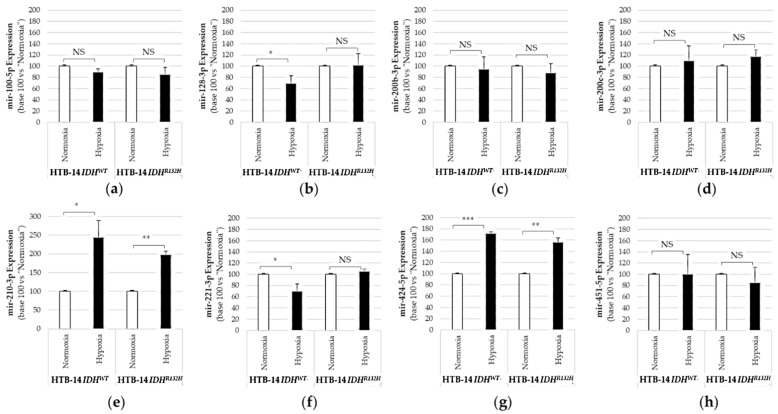
Expression of pro-angiogenic or pro-hypoxic miRNAs in HBT-14 (U-87 MG) *IDH1*^R132H^ or in HBT-14 (U-87 MG) *IDH1*^WT^ cells grown in hypoxia (1% O_2_, 24 h). HBT-14 (U-87 MG) *IDH1*^WT^ and HBT-14 (U-87 MG) *IDH1*^R132H^ cells at 60% confluence were cultivated for an additional 24 h in normoxia or hypoxia (0.1% O_2_). The miRNAs from HBT-14 (U-87 MG) *IDH1*^WT^ and HBT-14 (U-87 MG) *IDH1*^R132H^ cells were extracted using miRNAeasy (Qiagen™), then retrotranscribed (RT) and amplified (PCR) using the TaqMan MiRNA Reverse transcription kit (Applied Biosystem). The RT-PCR data were normalized to the small nucleolar house-keeping RNA, RNAS RNU48 (SNORD48) (assay ID 001006). Each miRNA ((**a**): mir-100-5p; (**b**): mir-128-3p; (**c**): mir-200b-3p; (**d**): mir-200c-3p; (**e**): mir-210-3p; (**f**): mir-221-3p; (**g**): mir-424-5p; (**h**): mir-451-5p) was thus finally expressed in base 100 (100 being attributed to the delta-CT of the miRNA measured in HBT-14 (U-87 MG) *IDH1*^WT^ or HBT-14 (U-87 MG) *IDH1*^R132H^ cells grown in normoxia) (*n* = 3, ANOVA followed by a post hoc Dunnett’s test, *: *p* < 0.05; **: *p* < 0.01; ***: *p* < 0.001, NS: non-significant).

**Figure 4 ijms-23-06042-f004:**
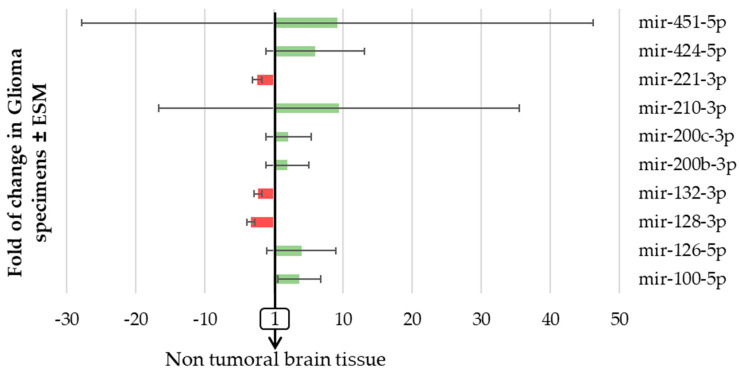
The expression of pro-angiogenic or pro-hypoxic miRNAs varied markedly in patients with glioma. miRNA from the 97 FFPE surgically resected tumor specimens or the 8 healthy brain tissues were extracted using miRNAeasy-FFPE kit (Qiagen™), then retrotranscribed (RT) and amplified (PCR) using the TaqMan MiRNA reverse transcription kit (Applied Biosystem). The RT-PCR data were normalized to the small nucleolar house-keeping RNA, RNAS RNU48 (SNORD48) (assay ID 001006). Results are expressed as the fold change in glioma samples ± ESM compared to healthy brain tissue.

**Figure 5 ijms-23-06042-f005:**
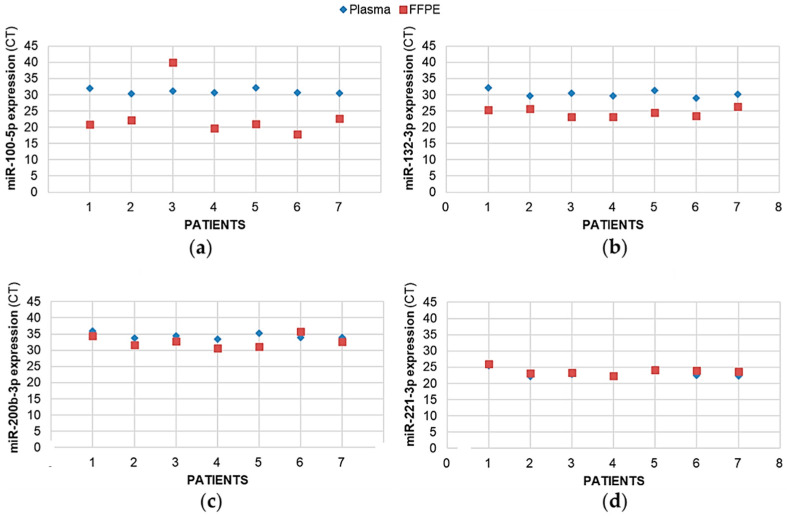
The expression levels of pro-angiogenic or pro-hypoxic miRNAs were correlated between the tumor sample and plasma from patients with glioma. For 7 patients with glioma, the miRNAs from the FFPE surgically resected tumor specimens or the plasma samples were extracted using, miRNAeasy-FFPE kit (Qiagen™) and NucleoSpin miRNA Plasma (Macherey-Nagel™), respectively, then retrotranscribed (RT) and amplified (PCR) using the TaqMan MiRNA Reverse transcription kit (Applied Biosystem). Results illustrated here for (**a**) mir-100-5p, (**b**) mir-132-3p, (**c**) mir-200b-3p, and (**d**) mir-221-3p are expressed as cycle thresholds (CTs) assayed for each miRNA normalized to the small nucleolar house-keeping RNA, RNAS RNU48 (SNORD48) (assay ID 001006).

**Table 1 ijms-23-06042-t001:** Inter-miRNA correlation (Spearman rank correlation coefficients).

	mir-200c-3p	mir-210-3p	mir-100-5p	mir-126-5p	mir-132-3p	mir-221-3p	mir-424-5p	mir-128-3p	mir-451-5p
mir-200b-3p	0.707	0.596	0.713	0.740	0.700	0.784	0.793	0.385	0.619
mir-200c-3p		0.494	0.591	0.778	0.726	0.791	0.764	0.355	0.641
mir-210-3p			0.435	0.550	0.344	0.543	0.549	0.078	0.464
mir-100-5p				0.642	0.627	0.607	0.741	0.535	0.514
mir-126-5p					0.729	0.797	0.781	0.434	0.749
mir-132-3p						0.822	0.725	0.592	0.546
mir-221-3p							0.728	0.381	0.621
mir-424-5p								0.549	0.702
mir-128-3p									0.539

Correlations were highly significant for all miRNAs (*p* < 0.001), except mir-128-3p and mir-210-3p (*p* = 0.45).

**Table 2 ijms-23-06042-t002:** MiRNA expression according to the *IDH1/2* mutation.

	*IDH1/2*-^WT^ (*n* = 37)			*IDH1/2*-^MUT^ (*n* = 60)			*p*
	Median (%)	First Quartile (%)	Third Quartile (%)	Median (%)	First Quartile (%)	Third Quartile (%)	
mir-200b-3p	250.5	82.9	450.0	59.3	21.8	97.8	<0.001
mir-200c-3p	262.4	122.0	449.0	64.8	20.9	138.9	<0.001
mir-210-3p	1109.0	380.1	1891.2	126.7	67.2	201.3	<0.001
mir-100-5p	362.9	189.1	597.7	238.6	141.2	410.4	0.086
mir-126-5p	451.0	175.7	1194.3	119.6	38.0	294.6	<0.001
mir-132-3p	32.1	18.4	56.0	20.0	9.2	36.7	0.024
mir-221-3p	35.7	17.4	90.4	8.5	3.0	18.5	<0.001
mir-424-5p	699.8	257.9	1324.0	215.4	104.8	410.5	<0.001
mir-128-3p	4.3	1.2	21.9	9.9	5.4	37.8	0.027
mir-451-5p	322.7	67.4	1104.3	73.5	25.9	310.6	0.0010

The 100 value was attributed to the miRNA expression in non-tumoral tissue. IDH: isocitrate dehydrogenase.

**Table 3 ijms-23-06042-t003:** MiRNA expression according to microvascular proliferation.

	No Microvascular Proliferation (*n* = 40)			Microvascular Proliferation (*n* = 57)			*p*
	Median (%)	First Quartile (%)	Third Quartile (%)	Median (%)	First Quartile (%)	Third Quartile (%)	
mir-200b-3p	62.0	28.6	115.0	99.0	26.0	386.4	0.021
mir-200c-3p	57.7	21.0	162.4	138.7	71.3	359.7	0.010
mir-210-3p	117.4	53.3	154.0	498.0	192.7	1568.4	<0.001
mir-100-5p	251.9	176.8	420.4	267.5	124.8	579.3	0.96
mir-126-5p	120.2	43.8	334.4	275.7	117.5	688.3	0.0073
mir-132-3p	26.4	9.5	42.3	25.5	13.0	49.3	0.51
mir-221-3p	8.6	3.8	20.6	19.5	7.8	63.2	0.0063
mir-424-5p	192.2	114.6	366.2	447.5	179.9	1075.5	0.0063
mir-128-3p	10.0	5.4	42.5	5.8	2.2	26.4	0.087
mir-451-5p	59.9	25.9	261.2	256.1	51.8	879.9	0.0068

The 100 value was attributed to the miR expression in non-tumoral tissue.

**Table 4 ijms-23-06042-t004:** MiRNA expression according to the WHO 2016 classification.

	O (*n* = 11)	AO (*n* = 16)	A (*n* = 18)	AA (*n* = 7)	GB-IDH^MUT^ (*n* = 8)	GB-IDH^WT^ (*n* = 37)	*p*
	Median (%)						
mir-200b-3p	37.9	35.0	58.9	96.6	63.5	250.5	<0.001
mir-200c-3p	45.0	88.9	66.7	166.1	53.0	262.4	<0.001
mir-210-3p	98.1	139.1	110.5	118.2	228.1	1109.0	<0.001
mir-100-5p	208.3	241.2	350.8	242.7	231.6	362.9	0.28
mir-126-5p	105.9	274.3	113.3	131.8	103.5	451.0	<0.001
mir-132-3p	15.6	18.6	23.1	47.8	15.2	32.1	0.10
mir-221-3p	4.6	8.1	8.5	13.6	8.8	35.7	<0.001
mir-424-5p	115.0	271.1	203.9	574.3	274.9	699.8	0.0012
mir-128-3p	7.1	12.9	15.2	9.7	10.6	4.3	0.28
mir-451-5p	71.9	313.9	53.9	58.2	116.3	322.7	0.0077

A value of 100 value was attributed to the miR expression in non-tumoral tissue. A: diffuse astrocytoma, IDH-mutant; AA: anaplastic astrocytoma, IDH-mutant; AO: anaplastic oligodendroglioma, IDH-mutant, and 1p19q-codeleted; GB-IDHMUT: glioblastoma, IDH-mutant; GB-IDHWT: glioblastoma, IDH-wild-type; IDH: isocitrate dehydrogenase; O: oligodendroglioma, IDH-mutant, and 1p19q-codeleted; WHO: World Health Organization.

**Table 5 ijms-23-06042-t005:** MiRNA expression and overall and progression-free survival of patients with glioma.

	HR ^†^	IC95%		*p*	Adjusted HR ^‡^	IC95%		*p*
OS								
mir-200b-3p	1.08	1.05	1.12	<0.001	1.03	0.98	1.08	0.30
mir-200c-3p	1.06	1.03	1.09	<0.001	1.02	0.98	1.05	0.42
mir-210-3p	1.015	1.008	1.02	<0.001	1.01	0.997	1.01	0.23
mir-100-5p	1.02	0.90	1.15	0.80	0.92	0.81	1.06	0.25
mir-126-5p	1.08	1.03	1.12	<0.001	0.99	0.94	1.05	0.83
mir-132-3p	1.04	1.00	1.08	0.055	1.02	0.98	1.07	0.38
mir-221-3p	1.07	1.04	1.11	<0.001	1.01	0.97	1.05	0.50
mir-424-5p	1.07	1.02	1.13	0.0038	1.00	0.95	1.06	0.94
mir-128-3p	1.02	0.97	1.07	0.40	1.00	0.96	1.04	0.99
mir-451-5p	1.008	1.003	1.01	0.0022	1.003	0.997	1.01	0.31
PFS								
mir-200b-3p	1.07	1.03	1.11	<0.001	1.02	0.96	1.07	0.56
mir-200c-3p	1.06	1.03	1.09	<0.001	1.02	0.98	1.06	0.44
mir-210-3p	1.013	1.007	1.02	<0.001	1.01	0.997	1.01	0.23
mir-100-5p	1.01	0.90	1.12	0.92	0.97	0.86	1.09	0.62
mir-126-5p	1.06	1.02	1.11	0.007	0.99	0.93	1.04	0.63
mir-132-3p	1.04	1.00	1.08	0.053	1.02	0.98	1.06	0.41
mir-221-3p	1.06	1.03	1.09	<0.001	1.00	0.97	1.04	0.90
mir-424-5p	1.07	1.02	1.12	0.0034	1.00	0.95	1.06	0.93
mir-128-3p	1.01	0.98	1.05	0.45	0.99	0.96	1.03	0.76
mir-451-5p	1.007	1.00	1.01	0.0077	1.003	0.996	1.01	0.40

^†^ mir-132-3p, mir-221-3p, and mir-128-3p for a 10-point increase in the level of expression; mir-200b-3p and mir-200c-3p for a 50-point increase in the level of expression; mir-210-3p, mir-126-5p, and mir-451-5p for a 100-point increase in the level of expression; mir-100-5p and mir-424-5p for a 150-point increase in the level of expression. ^‡^ Adjustments for sex, age at diagnosis, and WHO 2016 classification.

**Table 6 ijms-23-06042-t006:** Value of mir-100-5p or mir-128-3p within WHO 2016 classes for PFS.

	mir-100-5p				mir-128-3p			
	HR	IC95%		*p*	HR	IC95%		*p*
O	0.75	0.28	2.00	0.56	0.95	0.75	1.21	0.66
A	0.79	0.57	1.11	0.18	0.97	0.72	1.30	0.82
AO	0.99	0.776	1.26	0.94	1.33	1.046	1.68	0.020
AA	5.12	1.84	14.24	0.0018	1.31	1.06	1.60	0.010
GB-IDH^MUT^	0.68	0.42	1.08	0.10	0.93	0.65	1.31	0.66
GB-IDH^WT^	1.05	0.90	1.23	0.52	0.99	0.95	1.03	0.55

A: diffuse astrocytoma, IDH-mutant; AA: anaplastic astrocytoma, IDH-mutant; AO: anaplastic oligodendroglioma, IDH-mutant, and 1p19q-codeleted; GB-IDHMUT: glioblastoma, IDH-mutant; GB-IDHWT: glioblastoma, IDH-wild-type; IDH: isocitrate dehydrogenase; O: oligodendroglioma, IDH-mutant, and 1p19q-codeleted; WHO: World Health Organization.

## Data Availability

All data are stored at the CHU of Caen and CFB center and can be made available upon request.

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
