# Peer review of "The Use of Pro-Angiogenic and/or Pro-Hypoxic miRNAs as Tools to Monitor Patients with Diffuse Gliomas"

_ijms, 2022, doi:10.3390/ijms23116042_

Round 1
Reviewer 1 Report
The authors of, "The use of pro-angiogenic and/or pro-hypoxic miRNAs as tools to monitor patients with diffuse gliomas" have presented a study of miRNAs in glioma. Through both in vitro experiments and studies of archived glioma patient tissues, the authors have identified miRNAs that may be prognostic and perhaps serve as potential blood biomarkers. Although the authors have described the limitations of their findings based on the small sample sizes, the presented data are of interest for glioma researchers, and set the stage for future prospective work.
Author Response
We would like to thank the reviewer #1 for his/her interests and valuable comments on the manuscript.
As this reviewer foresees, we are currently in the process of forming a cohort, this time prospective, to determine whether the miRNAs identified in this article will be effective tools for monitoring patients with IDHMUT glioma.
Reviewer 2 Report
A brief summary:
The authors of this manuscript evaluated the expression of pro-angiogenic and/or pro hypoxic microRNAs (miRNAs) in both glioblastoma (GBM) in vitro and from tumoral surgical specimens embedded in the paraffin. First, the authors evaluated miRNA expression in GBM IDHwt and IDHmut cells comparing changes in hypoxic and normoxic conditions. Then, the expression of miRNAs was evaluated on GBM biopsy specimens, analyzing the differences according to the IDH mutation status, microvascular proliferation and histological subgroups of the WHO 2016 classification. In conclusion, the authors performed a prognostic evaluation of miRNAs identifying two of them involved in a lower progression free survival in relation to the IDH mutation status. Finally, miRNAs circulating in plasma have been also confirmed from patients affected by GBM.
Broad comments:
The study is interesting and shows a promising approach for the identification of miRNAs from a diagnostic and prognostic point of view, in patients affected by GBM. In the first part, the work is smooth and well structured; then, I suggest to describe data in a more exhaustive and complete manner for each figure, since the reader may lose the flow of work and the biological meaning about the results relating the evaluation of miRNA in samples derived from patients according to the mutational status of IDH, the microvascular proliferation, and the histological subgroup.
Specific comments:
- Figure 1: The expression of mir-126-5p is totally missing. Wouldn't it be more appropriate to exclude it from the figure?
- Figure 1: The significant increase of miRNAs in GBM IDHwt cells could be discussed in more detail. How their modulation may be involved in angiogenic and pro-hypoxic mechanisms?
- Figure 2: I suggest adding the scale bar in the phase contrast photos.
- Lines 136-138: the authors state “hypoxia modifies the expression of some of the miRNAs studied here in U87 cells depending on the existence of an IDH1 mutation”. I would change this sentence as the results show that the IDH mutation does not excessively affect the expression of miRNA in relation to hypoxic condition; the only differences are for mir-128-3p and mir-221-5p.
- I recommend improving the resolution and quality of figures where possible.
- U87 cells in the text are also referred as HTB14. Could the authors describe if there is a difference about the terminology or are they synonyms? In any case I suggest that the nomenclature should be uniformed in the text.
- Captions of figures repeatedly report the same redundant methodological procedures. It would be appropriate to describe the materials and methods in the proper section, avoiding repeating them also in the results of figures.
- Why did the authors not refer to the most updated WHO 2021 classification? Is it possible to justify this choice in the text?
- I suggest mentioning this recent review to discuss the correlations between the IDH status and the interconnection with the main hallmarks of GBM: doi: 10.3390/biomedicines10040806. PMID: 35453557; PMCID: PMC9031586
Author Response
Broad comments:
The study is interesting and shows a promising approach for the identification of miRNAs from a diagnostic and prognostic point of view, in patients affected by GBM. In the first part, the work is smooth and well structured; then, I suggest to describe data in a more exhaustive and complete manner for each figure, since the reader may lose the flow of work and the biological meaning about the results relating the evaluation of miRNA in samples derived from patients according to the mutational status of IDH, the microvascular proliferation, and the histological subgroup.
We would like to thank the reviewer#2 for his/her interests and valuable comments on the manuscript.
Thanks to the remark of the reviewer#2, we now explained in more exhaustive and complete manner each figure and table.
Specific comments:
Figure 1: The expression of mir-126-5p is totally missing. Wouldn't it be more appropriate to exclude it from the figure?
Thank you for this comment: we had considered excluding it but we did not want to give the impression that we had not assayed this mir-126-5p it in these lines. Indeed, if there is no value it is because this mir is undetectable in this cell line. We propose to this reviewer, to discard this mir from the Figure 1 and to comment in the text and legend that “the mir-126-5p was undetectable in these lines”.
Figure 1: The significant increase of miRNAs in GBM IDHwt cells could be discussed in more detail. How their modulation may be involved in angiogenic and pro-hypoxic mechanisms?
We thank the reviewer2 for this relevant suggestion, we thus now discussed this point as following in the discussion section: “Interestingly, induced-hypoxia of IDH HTB-14 (U-87 MG) cells more specifically modifies the expression of four miRNA, for two (mir-210-3p and mir-424-5p, which decrease under hypoxic condition) independently of IDH mutation, while for the two others (128-3p and mir-221-5p, which decrease under hypoxic condition) only in wild type IDH HTB-14 (U-87 MG) cells. Such result could explain why IDHWT glioma are more aggressive tumors and with a higher microvascular proliferation than IDHMUT glioma: indeed, the mir-128-3p is a tumor suppressor [37] and both the mir-128-3p and the mir-221-5p have sometimes anti-angiogenic behavior by targeting in particular respectively VEGFC [38] and the Hypoxia-inducible factor 1 alpha [39]”.
Figure 2: I suggest adding the scale bar in the phase contrast photos.
The correction is made.
Lines 136-138: the authors state “hypoxia modifies the expression of some of the miRNAs studied here in U87 cells depending on the existence of an IDH1 mutation”. I would change this sentence as the results show that the IDH mutation does not excessively affect the expression of miRNA in relation to hypoxic condition; the only differences are for mir-128-3p and mir-221-5p.
We thank the reviewer2 for this relevant remark, we rephrased this confused sentence by “Therefore, hypoxia modifies the expression of mir-210-3p and mir-424-5p studied here in HTB-14 (U-87 MG) cells regardless of IDH1/2 status, unlike the mir-128-3p and mir-221-5p whose variation in expression un-der hypoxia seems to be linked by the presence of a wild type IDH.“.
I recommend improving the resolution and quality of figures where possible.
The corrections have been made, we have in particular adjusted the sharpness of the figures.
U87 cells in the text are also referred as HTB14. Could the authors describe if there is a difference about the terminology or are they synonyms? In any case I suggest that the nomenclature should be uniformed in the text.
When we look at the ATCC site (https://www.atcc.org/products/htb-14), we can see that U87 and HTB14 are synonyms, we have chosen to keep the two names to allow everyone to easily identify the model we have used, we have however harmonized the names in the text of the manuscript and corrected the names to ensure that they are correct (U-87 MG instead of U87, and HTB-14 instead of HTB14).
Captions of figures repeatedly report the same redundant methodological procedures. It would be appropriate to describe the materials and methods in the proper section, avoiding repeating them also in the results of figures.
We are sorry for these redundancies but a figure must be able to be understood independently of the text according to the recommendations to the authors which obliges to make these repetitions.
Why did the authors not refer to the most updated WHO 2021 classification? Is it possible to justify this choice in the text?
We thank the reviewer for pointing this out. We are referring to the classification of 2016 and not to that of 2021, because the patients included in this work have already been the subject of a preliminary study. The tumor samples are for many of them well consumed, which no longer allows additional analyzes. Indeed, the cohort of patients presented here is a cohort on which we had already been able to ask the question of the role of alterations in the RASSF1A/Hippo signaling pathway in 2019 (PMID: 31055025 DOI: 10.1016/j.jmoldx.2019.03.007). The tumor samples had been reviewed in 2019 to be all requalified on the 2016 classification as explained in Levallet et al., 2019 (PMID: 31055025 DOI: 10.1016/j.jmoldx.2019.03.007). With i) the analyzes dues to the requalification of the tumor samples on the 2016 classification, ii) the DNA extractions having allowed the characterization of the hypermethylation of the promoters of the RASSF/Hippo genes from our 2019 study and iii) the extractions of the microRNAs necessary for the current study, the samples tumors of the patients were very consumed and it was no longer possible for us for many of them to redo the analyzes which would have made it possible to requalify the few patients who should have been reclassified according to the 2021 classification.
We are currently in the process of forming a prospective cohort, this time requalified on the 2021 classification, to determine whether the miRNAs identified in this article will be effective tools for monitoring patients with glioma.
As suggested by this reviewer, for the reader, we now explain this as following in the material and methods section: “All tumor specimens were reviewed by a neuropathologist (ELZ) to confirm the diagnosis and grade according to the new classification system adopted by the World Health Or-ganization (WHO) in 2016, as described in [10]. Indeed, we used the classification in place at the time of the constitution of this cohort, i.e. the WHO 2016 classification and could not update on the classification of 2021 due to the exhaustion of numerous tumor specimens.”.
I suggest mentioning this recent review to discuss the correlations between the IDH status and the interconnection with the main hallmarks of GBM: doi: 10.3390/biomedicines10040806. PMID: 35453557; PMCID: PMC9031586
We thank the reviewer for this suggestion, we now cited this review (please see, the new reference [1]) instead of the reference of Wang et al., 2018.